# Multispectral Optoacoustic Tomography: Intra- and Interobserver Variability Using a Clinical Hybrid Approach

**DOI:** 10.3390/jcm8010063

**Published:** 2019-01-09

**Authors:** Anne Helfen, Max Masthoff, Jing Claussen, Mirjam Gerwing, Walter Heindel, Vasilis Ntziachristos, Michel Eisenblätter, Michael Köhler, Moritz Wildgruber

**Affiliations:** 1Institute of Clinical Radiology, Medical Faculty, University of Muenster, and University Hospital Muenster, 48149 Muenster, Germany; anne.helfen@ukmuenster.de (A.H.); max.masthoff@ukmuenster.de (M.M.); Mirjam.Gerwing@ukmuenster.de (M.G.); heindel@uni-muenster.de (W.H.); eisenblaetter@uni-muenster.de (M.E.); michael.koehler@ukmuenster.de (M.K.); 2iThera Medical, 81379 Munich, Germany; jing.claussen@ithera-medical.com; 3Institute for Biological and Medical Imaging, Technical University of Munich, 81675 Munich, Germany; vntziachristos@gmail.com; 4Center for Translational Cancer Research ‘TranslaTUM’, Technical University of Munich, 81675 Munich, Germany; 5Division of Imaging Sciences & Biomedical Engineering, King’s College London, London SE1 7EH, UK; 6DFG EXC 1003 Cluster of Excellence ‘Cells in Motion’, University of Muenster, 48149 Muenster, Germany

**Keywords:** multispectral optoacoustic tomography, photoacoustic imaging, intrarater variability, interrater variability, reproducibility

## Abstract

Multispectral optoacoustic tomography (MSOT) represents a new imaging approach revealing functional tissue information without extrinsic contrast agents. Using a clinical combined ultrasound (US)/MSOT device, we investigated the interindividual robustness and impact of intra- and interobserver variability of MSOT values in soft tissue (muscle and subcutaneous fat) of healthy volunteers. Semiquantitative MSOT values for deoxygenated (Hb), oxygenated (HbO_2_) and total hemoglobin (HbT), as well as oxygen saturation (sO_2_), were calculated for both forearms in transversal and longitudinal probe orientation (*n* = 3, 8 measurements per subject). For intraobserver reproducibility, the same examiner investigated three subjects twice. Mean values of left vs. right forearm and transversal vs. longitudinal probe orientation were compared using an unpaired Student’s *t* test. Bland Altmann plots with 95% limits of agreement for absolute averages and differences were calculated. Intraclass correlation coefficients (ICC 2,k) were computed for three different examiners. We obtained reproducible and consistent MSOT values with small-to-moderate deviation for muscle and subcutaneous fat tissue. Probe orientation and body side had no impact on calculated MSOT values (*p* > 0.05 each). Intraobserver reproducibility revealed equable mean values with small-to-moderate deviation. For muscular tissue, good ICC was obtained for sO_2_. Measurements of subcutaneous tissue revealed good-to-excellent ICCs for all calculated values. Thus, in this preliminary study on healthy individuals, clinical MSOT provided consistent and reproducible functional soft tissue characterization, independent on the investigating personnel.

## 1. Introduction

In recent years, multispectral optoacoustic tomography (MSOT) has evolved as a new imaging approach, revealing functional tissue information non-invasively.

The MSOT principle is based on the photoacoustic effect: Thermoelastic expansion of target tissue, evoked by absorption of ultrashort laser pulses, is transmitted into ultrasound waves that can be detected and used for image reconstruction processes after spectral unmixing [1]. Due to specific absorber characteristics of intrinsic biomarkers like hemoglobin, melanin or lipids, distribution can be depicted quantitatively and with high spatial resolution within the tissue of interest. Furthermore, compared with other optical approaches, MSOT has a good tissue penetration, reaching up to a few centimeters [2,3] and enabling imaging in the clinical setting.

Recently, the feasibility of MSOT has been demonstrated in first clinical applications, revealing promising potential under different pathologic circumstances, such as cancer [4,5,6] and inflammatory diseases [7,8], as well as in physiologic vasculature [9,10]. In these studies, parameters like oxygenated or deoxygenated hemoglobin, as well as oxygen saturation, have been proven to be valuable biomarkers that characterize healthy and pathologic tissue.

The combination of ultrasound (US) and MSOT in one handheld device offers the advantage that—in addition to morphological tissue information depicted by US—functional properties can be simultaneously assessed and exactly co-localized [4,9], and different clinical handheld devices are currently being developed [11,12]. Due to its easy application and quick examination within the same timeframe as compared to conventional ultrasound, MSOT offers a high potential for clinical translation. However, since US is well known as highly dependent on the investigator’s experience [13,14], reproducibility studies regarding MSOT examination variability are urgently needed with upcoming clinical translation of the technique.

In this study we investigated the interindividual robustness, as well as the intra- and interrater variability, of MSOT imaging using a handheld probe of superficial soft tissue in healthy volunteers.

## 2. Materials and Methods

### 2.1. Study Design and Subjects

This preliminary study was separated into three different parts, each investigating superficial muscular and subcutaneous fat tissue of the left and right inner forearm to avoid artifacts caused by hairs, 15 cm proximal to the wrist, in the transversal and longitudinal probe orientation of healthy volunteers (4 measurements per healthy volunteer). As a first step, healthy volunteers (*n* = 3, 4 measurements each, 12 measurements in total, mean age 25.6 ± 2 year, age range: 23–28 years) were examined in regards to differences in body side (left vs. right forearm) and probe orientation (transversal vs. longitudinal), as well as to the robustness of calculated MSOT values. Second, the same healthy volunteers (*n* = 3, identical with the first step) were investigated by the same examiner a second time for evaluation of intraobserver variability/reproducibility. Concerning interobserver variability as a third step, MSOT of healthy volunteers (*n* = 4, mean age: 32.5 ± 12, age range: 23–53 years) was performed by 3 examiners each. Between different exams, the combined US/MSOT probe was placed on the device. The period between the individual examinations was more than 15 min each.

Investigations were carried out following the rules of the Declaration of Helsinki of 1975, revised in 2013. The study protocol has been reviewed by the local ethics committee (Ethikkommission der Westfälischen Wilhelms-Universität, Protocol No. 2017-538-f-S), and all volunteers gave written and informed consent prior to study enrolment. 

### 2.2. Technical Aspects of MSOT Imaging Device

This study was performed using a hybrid clinical MSOT/US imaging system equipped with a handheld probe (MSOT Acuity Echo, iThera Medical, Munich, Germany). The system consists of a tunable optical parametric oscillator (OPO), which is pumped by an Nd:YAG laser providing excitation pulses with a duration of 9 at wavelengths from 680 to 980 nm. The probe was connected to the OPO via a fiber bundle integrated into the probe and a diffuser providing an elliptical light spot of approximately 10 mm width and 15 mm length.

With respect to the American National Standards Institute limits of maximum permissible exposure, the pulse energy was attenuated to a maximum of 30 mJ at 750 nm. Multispectral images were acquired using 1 pulse per wavelength image. The detector (256 transducer elements of center frequency = 4 MHz; send/receive bandwidth = 52%, aperture size transducer 4 MHz) had a 135° angular coverage providing 2D cross-sectional images with a field of view of 30 × 30 mm^2^ and 100 μm reconstructed pixel size. Multispectral imaging was possible at up to 25 Hz with a 5 Hz refresh rate per multispectral image (25 Hz/5 wavelengths). Further technical details of the system have been previously described in detail elsewhere [4,7,8]. Penetration depth of the laser system in the current setup was limited to 3 cm, sufficient for imaging subcutaneous fat and superficial muscular tissue (Figure 1a). Reflection ultrasound computed tomography mode images were generated accordingly.

### 2.3. MSOT Image Acquisition

Each investigation period was about 10 min per volunteer. Eyes of examiners and patients were protected with laser safety goggles. Examiners were experienced in both ultrasound and optoacoustic imaging. Scans were performed under equal conditions with identical body positioning and a 15 min rest period of the probands before starting the examination at room temperature. The acquired 5 wavelengths for MSOT images were 700, 730, 760, 790 and 850 nm. After spectral unmixing, individual semiquantitative values of oxygenated (HbO_2_), deoxygenated (Hb) and total hemoglobin (HbT = HbO_2_ + Hb), as well as oxygen saturation (sO_2_), were calculated from the acquired data of superficial soft tissue. Subsequently, these MSOT parameters were pseudocolor-coded and visualized with the US as composite images.

### 2.4. Data Analysis and Statistics

Data were analyzed using cLabs software (iThera, Munich, Germany) and MATLAB (VersionR2017b, TheMathWorks, Inc., Natick, MA, USA).

Due to the ultrasound co-localization of the region of interest (ROI), identification was highly standardized and performed in consensus of all examiners by defining definite ROIs, based on the ultrasound images, separately for subcutaneous tissue at 5 mm depth (50 mm²) and for muscular tissue at 10 mm depth (100 mm², Figure 1a). Results are indicated in arbitrary units (a.u.). 

Statistical analysis was performed using GraphPad Prism (version 7, GraphPad Software Inc.). Mean values and standard deviation were calculated for all individual MSOT parameters (Hb, HbO_2_, HbT and sO_2_). For differences in body-side and probe orientation, an unpaired Student’s *t* test was used for statistical analysis. *p* values < 0.05 were considered to be significant.

For intraobserver variability/reproducibility, Bland Altman plots with 95 % limits of agreement for absolute average differences were created using GraphPad Prism in addition to mean values and standard deviation. Intraclass correlation coefficients (ICC 2,k, two-way random average measures, consistency/absolute agreement) were calculated using IBM SPSS Statistics (version 25, International Business Machines Corporation) for interobserver variability. ICC values from 0.41 to 0.60 were considered to correspond with moderate, from 0.61 to 0.80 with substantial/good and from 0.81 to 1.00 with excellent agreement [15].

## 3. Results

Non-invasive, contrast-agent free real-time hybrid MSOT/US imaging allowed for an exact anatomical localization of superficial soft tissue and muscle in reconstructed data, followed by quantification of spectrally unmixed signals for hemoglobin and oxygen saturation quantification (Figure 1b). 

In all measurements (*n* = 12) of the first examination of healthy volunteers (*n* = 3), no significant differences could be detected between left and right forearms or transversal and longitudinal probe orientation for Hb, HbO_2_, HbT and sO_2_ in muscular and subcutaneous tissue (*p* > 0.05, detailed values are presented in Table 1). 

Calculated values of the first and second examination of both sides, as well as both orientations (*n* = 24), were therefore pooled with regard to both muscular and subcutaneous tissue in subsequent experiments. For both muscular (Figure 1c) and subcutaneous tissue (Figure 1d) MSOT revealed signals for calculated hemoglobin values as well as sO_2_ with small-to-moderate scattering between single measurements. 

As expected, mean values for Hb and HbO_2_ were significantly higher in muscular tissue compared with subcutaneous tissue due to subcutaneous tissue’s composition of fat and connective tissue, as well as its possession of a smaller number of blood vessels (40.52 ± 5.3 vs. 25.94 ± 3.91 a.u., *p* < 0.0001; and 34.48 ± 7.83 vs. 29.61 ± 7.86 a.u., *p* = 0.0368). The smallest standard deviations were observed for sO_2_ in muscular tissue and Hb in subcutaneous tissue (Table 1). 

Mean values and standard deviation of data regarding intraobserver variability/reproducibility are presented in Table 2. Calculated data revealed a good reproducibility for repetitive investigation of the same examiner, especially regarding sO_2_ values of muscle (bias of 0.44 ± 3.75 a.u., 95 % limits of agreement between –6.92 and 7.81 a.u.) and subcutaneuous fat (bias of 3.4 ± 6.39 a.u., 95 % limits of agreement between –9.12 and 15.92 a.u.). In terms of absolute differences and averages, calculated values did not show a large dispersion for either muscular (Figure 2a) or subcutaneous tissue (Figure 2b) in Bland Altman analyses.

ICC calculation for interobserver variability revealed moderate agreement for Hb, HbO_2_ and HbT in muscular tissue, but good (ICC = 0.69) for sO_2_. However, in subcutaneous tissue acquired, MSOT values showed good correlation coefficients (ICC = 0.72) for sO_2_, and excellent correlation coefficients for Hb (ICC = 0.86) and HbT (ICC = 0.81, Table 3).

## 4. Discussion

In this pilot study, we presented first data concerning the interindividual comparability of MSOT values acquired by a handheld system in healthy superficial soft tissue. We aimed to study data acquisition dependent intra- and interobserver variability. Data proved to be stable and interindividually comparable with only little variances in relation to the intrinsic biomarkers hemoglobin and sO_2_ in both muscular and subcutaneous tissue (Table 1, Figure 1c,d). Overall, values showed more spread in muscles as compared to subcutaneous fat, which is presumably caused by interindividual differences in previous muscular activity with concomitant increased tissue perfusion or accidentally captured intramuscular blood vessels of large diameters. Furthermore, with increasing penetration depth of deeper muscle tissue, light scattering and signal loss may increase limiting spectral unmixing algorithms.

Examinations were performed by trained examiners in both ultrasound and MSOT. However, there may be little interindividual variations concerning the positioning of the probe and the resulting angle of the incident light due to movements of the probe. Nevertheless, mean values of healthy individuals may serve as standard/baseline for comparison with pathologically altered biomarkers.

Repetitive investigation by the same examiner revealed good reproducibility of MSOT values, indicating low intraobserver variability. Furthermore, interclass correlation coefficients revealed moderate-to-good agreement for muscular, and good-to-excellent agreement for subcutaneous fat tissue. 

In recent years, the number of personalized diagnostics and therapies has increased steadily. Current clinically established imaging methods allow for the assessment of pathological parameters usually based on mere morphology. Functional assessment of tissue alterations by novel imaging biomarkers, occurring during early therapy response and especially prior to morphological changes, would therefore be highly desirable.

MSOT enables non-invasive imaging of functional tissue biomarkers, which are expected to differ between physiologic and pathologic conditions without the need of extrinsic contrast agents. Besides its application in preclinical models of disease [16,17], MSOT has recently proven to be a promising tool in first clinical applications using intrinsic biomarkers such as hemoglobin or melanin [4,7,8,18]. Due to its similarities in handling with clinical established ultrasound, a handheld device of hybrid US/MSOT allows for easy use, even at the patient´s bedside. Scattering effects, which increasingly occur with growing tissue depth [4], were less considerable concerning this technical evaluation selecting the same tissue depth. However, penetration depth of the laser is currently limited to 3 cm, and the regions of interest were placed on artifact-reduced hairless light skin types, requiring further technical developments and studies for a broad clinical application. 

## 5. Conclusions

In summary, our preliminary data indicate consistent and reproducible functional soft tissue characterization, independent from the investigating personnel. In future trials confirmation of these data could therefore also be realized in clinical multicenter studies with enlarged study groups.

## Figures and Tables

**Figure 1 jcm-08-00063-f001:**
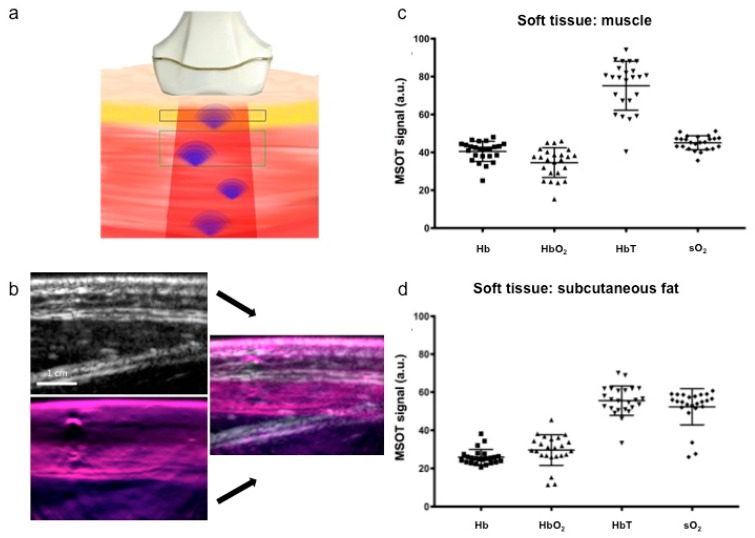
(**a**) MSOT principle: Based on the photoacoustic effect, laser pulses cause thermoelastic expansion within the tissue of interest followed by emission of detectable ultrasound waves. Light source and detector are combined in one probe. Due to different absorber properties, spectral unmixing enables quantitative analysis of the intrinsic biomarkers hemoglobin and oxygen saturation non-invasively. Regions of interest were placed within the muscular (green rectangle) and the subcutaneous fat tissue (blue rectangle). (**b**) Exemplary ultrasound (top left, scale: 1 cm) and pseudocolor-coded MSOT signals (bottom left) are merged into an overlay image (right). Acquired MSOT values of deoxygenated (Hb), oxygenated (HbO_2_) and total hemoglobin (HbT) as well as oxygen saturation (*n* = 24 each) reveal small to moderate deviation in muscular (**c**) and subcutaneous fat tissue (**d**).

**Figure 2 jcm-08-00063-f002:**
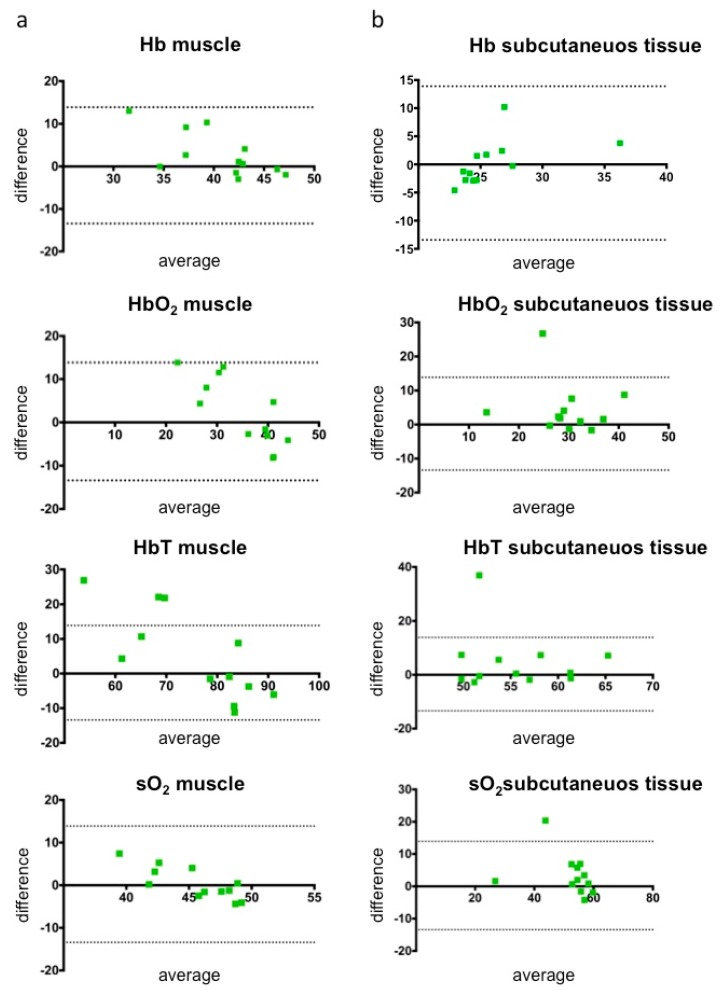
Bland Altman Plots illustrating absolute differences and averages between the mean values of first and second measurements acquired by the same examiner in healthy individuals (*n* = 3, 12 measurements, 95% limits of agreement are indicated by the dashed lines). For deoxygenated (Hb), oxygenated (HbO_2_) and total hemoglobin (HbT), as well as oxygen saturation in muscular (**a**) and subcutaneous fat tissue (**b**), good reproducibility of the calculated values can be detected, indicating a low intraobserver variability.

**Table 1 jcm-08-00063-t001:** MSOT mean values (a.u.) for muscular and subcutaneous tissue comparing both forearms as well as transversal and longitudinal probe orientation.

	Right Forearm		Left Forearm			Transversal Measurement		Longitudinal Measurement		
Muscle	Mean (*n* = 12)	SD	Mean (*n* = 12)	SD	*p*-Value	Mean (*n* = 12)	SD	Mean (*n* = 12)	SD	*p*-Value
Hb	40.64	5.08	40.43	5.54	0.92	42.20	4.49	38.88	5.56	0.12
HbO_2_	35.94	6.64	33.01	7.81	0.33	37.52	6.94	32.64	7.99	0.12
HbT	75.54	12.16	75.70	13.73	0.98	79.72	11.16	71.52	13.35	0.12
sO_2_	45.37	2.93	45.63	4.17	0.86	46.73	2.85	44.26	3.86	0.09
Subcutaneous Tissue	Mean (*n* = 12)	SD	Mean (*n* = 12)	SD	*p*-Value	Mean (*n* = 12)	SD	Mean (*n* = 12)	SD	*p*-Value
Hb	26.51	4.56	25.37	3.01	0.48	25.32	1.73	26.52	5.09	0.45
HbO_2_	29.01	8.44	30.21	7.18	0.71	29.73	3.20	29.49	10.64	0.94
HbT	55.52	5.52	55.58	9.11	0.98	55.05	4.31	56.05	9.72	0.75
sO_2_	51.12	11.05	53.54	6.88	0.53	54.50	2.59	50.16	12.50	0.25

**Table 2 jcm-08-00063-t002:** Intrarater variability of calculated MSOT values (a.u.): Comparison of first and second measurement acquired by the same examiner investigating three different volunteers.

	Proband 1				Proband 2				Proband 3			
	First Measurement		Second Measurement		First Measurement		Second Measurement		First Measurement		Second Measurement	
Muscle	Mean (*n* = 4)	SD	Mean (*n* = 4)	SD	Mean (*n* = 4)	SD	Mean (*n* = 4)	SD	Mean (*n* = 4)	SD	Mean (*n* = 4)	SD
Hb	40.54	3.8	40.64	3.61	40.72	2.60	31.92	4.14	44.61	1.48	44.81	2.64
HbO_2_	36.55	5.17	38.33	8.41	33.74	3.37	22.16	3.97	38.41	2.50	41.29	3.06
HbT	77.09	8.95	78.97	12.01	74.46	5.73	54.07	7.97	83.02	3.71	86.09	5.56
sO_2_	46.65	1.88	47.86	4.26	44.40	2.05	40.16	2.81	46.10	1.19	47.80	0.80
Subcutaneous Tissue	Mean (*n* = 4)	SD	Mean (*n* = 4)	SD	Mean (*n* = 4)	SD	Mean (*n* = 4)	SD	Mean (*n* = 4)	SD	Mean (*n* = 4)	SD
Hb	28.85	5.65	27.59	3.93	27.08	3.20	24.61	2.12	22.32	1.04	25.18	0.65
HbO_2_	29.06	8.44	28.50	9.89	37.27	5.37	25.49	9.08	29.27	2.40	28.07	2.21
HbT	57.91	3.39	56.09	6.30	64.35	5.51	50.10	10.48	51.59	2.74	53.25	2.80

**Table 3 jcm-08-00063-t003:** Interclass correlation coefficients (ICC) calculated from mean values for Hb, HbO_2_, HbT and sO_2_ (a.u.) for four healthy volunteers investigated by three different examiners each. ICCs from 0.41 to 0.60 were considered to correspond with moderate, from 0.61 to 0.80 with substantial/good and from 0.81 to 1.00 with excellent agreement.

Muscle	Subcutaneous Tissue
Hb
	Examiner 1	Examiner 2	Examiner 3		Examiner 1	Examiner 2	Examiner 3
Proband 1	38.55	41.61	48.22	Proband 1	23.99	24.26	25.64
Proband 2	35.49	35.94	38.07	Proband 2	25.68	28.70	26.36
Proband 3	36.73	35.84	32.54	Proband 3	18.21	19.52	14.83
Proband 4	40.52	37.66	39.56	Proband 4	23.06	21.60	22.91
ICC	0.50			ICC	0.86		
HbO_2_
	Examiner 1	Examiner 2	Examiner 3		Examiner 1	Examiner 2	Examiner 3
Proband 1	34.28	41.15	46.47	Proband 1	27.15	30.21	28.95
Proband 2	26.17	32.26	28.31	Proband 2	29.56	29.22	30.01
Proband 3	35.48	37.87	31.26	Proband 3	25.18	27.47	22.10
Proband 4	38.28	34.08	34.95	Proband 4	29.68	29.15	28.46
ICC	0.52			ICC	0.60		
HbT
	Examiner 1	Examiner 2	Examiner 3		Examiner 1	Examiner 2	Examiner 3
Proband 1	72.82	82.76	94.69	Proband 1	51.14	54.48	54.59
Proband 2	61.66	68.19	66.39	Proband 2	55.24	57.93	56.38
Proband 3	72.21	73.71	63.80	Proband 3	43.38	47.00	36.93
Proband 4	78.80	71.74	74.51	Proband 4	52.74	50.75	51.37
ICC	0.49			ICC	0.81		
sO_2_
	Examiner 1	Examiner 2	Examiner 3		Examiner 1	Examiner 2	Examiner 3
Proband 1	47.06	49.78	49.04	Proband 1	53.67	56.22	53.73
Proband 2	42.32	46.91	42.56	Proband 2	52.97	50.27	53.41
Proband 3	49.02	51.47	49.15	Proband 3	59.32	59.29	60.68
Proband 4	48.05	47.60	46.66	Proband 4	56.25	58.65	53.06
ICC	0.69			ICC	0.72

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
