# Peer review of "Multispectral Optoacoustic Tomography: Intra- and Interobserver Variability Using a Clinical Hybrid Approach"

_jcm, 2019, doi:10.3390/jcm8010063_

Reviewer 1 Report

This paper tried to evaluate the intraobserver and interobserver variability of the MSOT used to differentiate different tissues in the inner forearms. However, the reviewer would like to raise some points for the authors to further clarify:

The technical details were largely missing from the manuscript. For example, in the abstract, what statistical test(s) was used to compare the MSOT values should be addressed. Although the abstract should be concise, however, the necessary information should be in place so as to make the abstract more informative. 

Because the intraobserver variability was conducted with only two measurements, how could the authors evaluate the variability in this sense? 

In the method section, how the measures were conducted is unclear. Two orientations and two hands formed a 2 by 2 factorial design. Does it mean each orientation x each hand was measured twice in the experiment? As a result, 8 measurements were conducted for each subject. 

For the repeated measures, how could the authors make sure that the same location was measured for each of the two measurements? 

It is also not clear what the observers were really observing. Each of them specified different tissue types in the acquired images and then the corresponding physiological values, such as the SO2, etc, were used to find the consistency of the rating? This is really a key issue but I didn't see this being addressed in the entire scope of the manuscript. 

Author Response

Honored members of the Editorial Board, dear Reviewers,

we would like to express our sincere gratitude to the very valuable points raised by the reviewers. All comments were carefully considered and appropriate changes and amendments to the manuscript have been made. Every point made by the reviewers has been worked on and we provide a detailed point-by-point reply below. All changes carried out in the manuscript have been highlighted using the track changes mode. We believe that by carrying out those changes the quality of the manuscript improved significantly.

We hope, that the presented manuscript is now acceptable for publication as “brief report” in The Journal of Clinical Medicine.

Reviewer #1:

This paper tried to evaluate the intraobserver and interobserver variability of the MSOT used to differentiate different tissues in the inner forearms. However, the reviewer would like to raise some points for the authors to further clarify:

1) The technical details were largely missing from the manuscript. For example, in the abstract, what statistical test(s) was used to compare the MSOT values should be addressed. Although the abstract should be concise, however, the necessary information should be in place so as to make the abstract more informative.

RE: Thank the reviewer very much for this comment. We added the statistical tests to the abstract section to increase its informative content. Besides, we inserted further technical details on the MSOT system in the Material and Methods section under point 2.2.

2) Because the intraobserver variability was conducted with only two measurements, how could the authors evaluate the variability in this sense?

RE: We are aware of the small number of healthy volunteers. Although every healthy volunteer has been examined by two investigators, a sample size of 4 measurements has been acquired per healthy volunteer. Analysis of intraobserver variability as shown in the Bland Altman Plots therefore refers to 12 data points regarding the difference between measurement 1 and 2. Thus, due to the small data set in this pilot study, results were presented descriptively and standard deviations listed in Table 2, as recommended by a professional statistician from our institution.

3) In the method section, how the measures were conducted is unclear. Two orientations and two hands formed a 2 by 2 factorial design. Does it mean each orientation x each hand was measured twice in the experiment? As a result, 8 measurements were conducted for each subject.

RE: We thank the reviewer for this valuable comment. The Material and Methods section was indeed misleading at this point. Both forearms of healthy volunteers were examined in transversal and longitudinal probe orientation, resulting in 4 measurements per volunteer. Because there were no differences between body side and probe orientation (as shown in Table 1) and we performed MSOT twice in 3 healthy volunteers, 24 data points were pooled for Figure 1 c and 1d.

We have clarified this point both in the Methods and Results section.

4) For the repeated measures, how could the authors make sure that the same location was measured for each of the two measurements?

RE: Scans were performed in identical probe orientation on both forearms, 15 cm proximal to the wrist, in transversal and longitudinal probe orientation of healthy volunteers. The probe position was marked on the forearms to allow for similar probe position.

5) It is also not clear what the observers were really observing. Each of them

specified different tissue types in the acquired images and then the corresponding physiological values, such as the SO2, etc, were used to find the consistency of the rating? This is really a key issue but I didn't see this being addressed in the entire scope of the manuscript.

RE: ROIs were drawn in consensus of all examiners based on the ultrasound images separately for subcutaneous fat tissue at 5 mm depth (50 mm²) and for muscular tissue at 10mm depth (100 mm², Figure 1a). We stayed this now clearly under section 2.4.

Reviewer 2 Report

The manuscript submitted to Journal of clinical medicine is a brief report entitled “Multispectral Optoacoustic Tomography: Intra- and Interobserver Variability Using a Clinical Hybrid Approach” by Helfen et al., is an interesting research paper. In this brief report authors attempted to evaluate the biomarker status of (e.g., hemoglobin) using the Multispectral Optoacoustic Tomography (MSOT)/US imaging techniques without using external tracers by non-invasively. Author’s analyzed two sets of live tissues of muscle and subcutaneous fat. Analysis were performed by three examiners in four different volunteers. Overall the preliminary study on healthy individuals using clinical MSOT data indicated, a) reproducible mean value from intraobserver, and good ICC value for sO2 from muscle analysis. This report is well written and related data are presented nicely. Although, this is preliminary report with limited N value, but it will be useful for readers of imaging scientists.

Minor comments:

a) The semi quantitative MSOT values of the following markers indicating what pathological conditions? deoxygenated hemoglobin (Hb), oxygenated (HbO2) and total hemoglobin (HbT) as well as oxygen saturation (sO2). Authors need to indicate why they performing this intrinsic imaging study..?

b) In the discussion section, para #1 authors need to indicate the table/figure numbers in the following sentences for cross referencing data “Data proved to be stable and interindividually comparable with only little variances in relation to the intrinsic biomarkers hemoglobin and sO2 in both muscular and subcutaneous tissue. Overall, values showed more spread in muscles as compared to subcutaneous fat…”

Author Response

Honored members of the Editorial Board, dear Reviewers,

we would like to express our sincere gratitude to the very valuable points raised by the reviewers. All comments were carefully considered and appropriate changes and amendments to the manuscript have been made. Every point made by the reviewers has been worked on and we provide a detailed point-by-point reply below. All changes carried out in the manuscript have been highlighted using the track changes mode. We believe that by carrying out those changes the quality of the manuscript improved significantly.

We hope, that the presented manuscript is now acceptable for publication as “brief report” in The Journal of Clinical Medicine.          

Reviewer #2:

The manuscript submitted to Journal of clinical medicine is a brief report entitled “Multispectral Optoacoustic Tomography: Intra- and Interobserver Variability Using a Clinical Hybrid Approach” by Helfen et al., is an interesting research paper. In this brief report authors attempted to evaluate the biomarker status of (e.g., hemoglobin) using the Multispectral Optoacoustic Tomography (MSOT)/US imaging techniques without using external tracers by non-invasively. Author’s analyzed two sets of live tissues of muscle and subcutaneous fat. Analysis were performed by three examiners in four different volunteers. Overall the preliminary study on healthy individuals using clinical MSOT data indicated, a) reproducible mean value from intraobserver,

and good ICC value for sO2 from muscle analysis. This report is well written and related data are presented nicely. Although, this is preliminary report with limited N value, but it will be useful for readers of imaging scientists.

Minor comments:

a)    The semi quantitative MSOT values of the following markers indicating what pathological conditions? deoxygenated hemoglobin (Hb), oxygenated (HbO2) and total hemoglobin (HbT) as well as oxygen saturation (sO2). Authors need to indicate why they performing this intrinsic imaging study..?

RE: We thank the reviewer for this comment. Indeed, intrinsic biomarkers like oxygenated and deoxygenated hemoglobin were used to characterize healthy and pathologic tissue in first clinical applications and are therefore of high interest with regards of translating MSOT to clinics.

We added this point to the introduction and the discussion section of the manuscript.

b)    In the discussion section, para #1 authors need to indicate the table/figure numbers in the following sentences for cross referencing data “Data proved to be stable and interindividually comparable with only little variances in relation to the intrinsic biomarkers hemoglobin and sO2 in both muscular and subcutaneous tissue. Overall, values showed more spread in muscles as compared to subcutaneous fat…”

RE: We added the referring table and figure numbers to the indicated sentence.

Round  2

Reviewer 1 Report

Although the authors pointed out in their responses to the reviewers' comments, I could not see the corresponding revision in the context of the newly revised proposal. Please resubmit the revised version with highlighted portions for the corresponding recision clearly. Thanks.

Author Response

Dear Sirs,

we regret that obviously the changes made to the manuscript were not visible to the reviewer (we used the track changes mode in MS word, obviously there were problems associated with).

We now highlighted again all changes made by applying yellow color, and we hope that all changes are now clearly visible.

Please contact us if further qurestions/points remain.

Sincerley yours,

Moritz Wildgruber on behalf of the authors

Round  3

Reviewer 1 Report

Accepted as is.